# In-Situ Synthesis of Methyl Cellulose Film Decorated with Silver Nanoparticles as a Flexible Surface-Enhanced Raman Substrate for the Rapid Detection of Pesticide Residues in Fruits and Vegetables

**DOI:** 10.3390/ma14195750

**Published:** 2021-10-01

**Authors:** Qijia Zhang, Guangda Xu, Na Guo, Tongtong Wang, Peng Song, Lixin Xia

**Affiliations:** 1College of Chemistry, Liaoning University, Shenyang 110036, China; zqj13940305120@163.com (Q.Z.); xuguangdalnu@163.com (G.X.); 18404713138@163.com (N.G.); 18340312652@163.com (T.W.); 2Yingkou Institute of Technology, Yingkou 115014, China

**Keywords:** surface enhanced Raman scattering, flexible, thiram, MC/Ag NPs film, in situ

## Abstract

The purpose of this study was to develop a flexible substrate methylcellulose-decorated silver nanoparticles (MC/Ag NPs) film and explore its application in fruits and vegetables by surface enhanced Raman spectroscopy (SERS) technology for rapid detection of pesticides. The performance of the MC/Ag NPs film substrate was characterized by Nile blue A (NBA), and the detection limit was as low as 10^−8^ M. The substrate also exhibited satisfactory Raman signal strength after two months of storage. The impressive sensitivity and stability were due to the excellent homogeneity of the silver nanoparticles that were grown in situ in the methylcellulose matrix, which generated “hot spots” between the silver nanoparticles without a large amount of aggregation, and resulted in the ultra-high sensitivity and excellent stability of the MC/Ag NPs film substrate. The MC/Ag NPs film substrate was used to detect thiram pesticides on tomato and cucumber peels, and the minimum detectable level of thiram was 2.4 ng/cm^2^, which was much lower than the maximum residue level. These results indicate that the MC/Ag NPs film is sensitive to rapid detection of multiple pesticides in food.

## 1. Introduction

Pesticide residues in foods have attracted much attention in recent years as consuming fruits and vegetables that are contaminated with pesticides can cause potential health problems, such as poisoning [1]. Therefore, there is strong demand for the detection of pesticide residues in agricultural products. Common methods of analyzing pesticide residues include high performance liquid chromatography (HPLC) [2] and gas chromatography (GC) combined with different detection techniques [3]. Such methods are sensitive, standardized, and are common, and are regarded as the gold standard [4] for pesticide detection. However, these analytical methods also have some drawbacks, such as complex pre-processing steps, time-consuming analytical processes, professional technicians, and the need for large and expensive instrumentation. In recent years, surface enhanced Raman spectroscopy (SERS) has been regarded as a promising method for rapid and nondestructive sample detection. SERS technology has the ability to accurately identify the signal fingerprints of analytes, and it can be applied to multiple fields.

Since its discovery in the 1970s, surface enhanced Raman scattering (SERS) has attracted increasing interest as a powerful analytical tool that can provide vibrational spectral information for the identification and quantification of multiple analyses [5,6]. With the great development of laser technologies and nanotechnologies, the application of SERS has been extended to multiple fields, including agricultural [7,8,9,10], environmental [11], medical [12], and biological [13,14], and it has also been used for art protection [15]. To better apply SERS to such fields, an important requirement was to develop reasonably designed ultra-sensitive, highly stable and replicable SERS substrates that could generate strong local electromagnetic fields (EM) in the gap of plasma nanostructure, that is, SERS “hot spots” [16,17,18,19]. Traditional SERS substrate was usually composed of noble metals, such as silver, gold, or copper, which can be prepared as colloidal nanoparticles or rigid materials as substrates. Colloidal nanoparticles are generally cheap to prepare and easy to modify. Many functional chemicals have been used to modify such nanoparticles [20], but colloidal systems have serious aggregation problems. In addition, The SERS spectra collected from the aggregated colloidal system were determined by the experimental conditions [21]. In contrast, rigid substrates ensure that these colloidal structures have excellent uniformity and reproducibility. However, due to the difficulty in collecting surface analysis of irregular samples on rigid substrates, SERS detection often required complex sample pretreatment steps [22]. In addition, the application of rigid substrates was limited by their high costs and complicated manufacturing process. Therefore, it is still necessary to develop a multifunctional SERS substrate that is suitable for routine analysis.

To overcome the above challenges, scientists have recently studied a variety of flexible materials, such as polydimethylsiloxane (PDMS) [23], polymethyl methacrylate (PMMA) [24], chitosan [25], tape [26], and bacterial nanocellulose [27], among others, as the carrier materials for decorative precious metal nanoparticles. For example, Kumar et al. first prepared oriented silver nanorods using the swept angle deposition technique and the thermal evaporation of silver powder on a silicon substrate, after which they used a PDMS film [28] to strip the arrays from silicon wafers. Chen et al. constructed SERS tapes using the drop deposition method to decorate commercial tapes with Au NPs [26]. The constructed SERS tape could directly extract thiram pesticide from foods for SERS detection. In addition, a simple method for the in situ synthesis of various nanostructures on cellulose paper has also been reported [29]. Although these flexible substrates have many advantages in actual sample detection compared to traditional rigid substrates, they also have disadvantages such as the necessity for cumbersome manufacturing processes, inhomogeneous distribution of “hot spots”, and unstable SERS signals, which can limit their actual or on-site detection. As silver nanostructures stored in the air environment for a long time are easily oxidized, SERS enhancement effect will inevitably decrease during this process. Many methods have been evaluated for maintaining SERS performance of substrates, including coating the nanostructure, storing at low temperatures, or protecting with inert gases. Methyl cellulose is rich in hydroxyl and can be used as a carrier for the growth of nanoparticles. Moreover, it is easy to obtain, non-toxic, water-soluble, and cheap. Methylcellulose is a typical non-ionic polymer, which can stabilize ion concentration extremely well [30].

In this study, we developed a simple strategy to maintain the SERS activity of a substrate using the in situ growth of silver nanoparticles in a methylcellulose matrix as a film (MC/Ag NPs film). The matrix can protect the silver nanoparticles from the external environment. The substrate can not only be directly immersed in Nile blue A (NBA) solution for Raman detection, but can also be wiped on the surface of fruits and vegetables to detect pesticide residues. The MC/Ag NPs film substrate exhibited high detection sensitivity in the NBA solution at concentrations as low as 10^−8^ M. The MC/Ag NPs film substrate also exhibited high sensitivity for the detection of thiram pesticides. More importantly, the MC/Ag NPs film SERS can realize the complete extraction and detection process without any other conditions. Thus, the proposed flexible MC/Ag NPs film SERS substrate can be broadly applied in the field for food safety analyses.

## 2. Materials and Methods

### 2.1. Materials

Methyl cellulose (MC), Nile Blue A (NBA), and thiram pesticides were purchased from Aladdin Industrial Co. (Shanghai, China). Silver nitrate (AgNO_3_) was obtained from Sinopharm Chemical Reagent Co., Ltd. (Shanghai, China). Foods such as tomatoes and cucumbers were bought from a local supermarket in Shenyang, China. All aqueous solutions used in the study were freshly prepared in Milli-Q water (>18 MΩ·cm^−1^).

### 2.2. Characterizations

Scanning electron microscopic characterization was performed using a JEOL JSM-7400F Scanning electron microscope (SEM) at an operating voltage of 5 kV (Hitachi, Tokyo, Japan). Scanning electron microscope X-ray energy dispersive microanalysis (SEM-EDX) was performed using a Gemini 500 instrument (Zeiss, Oberkochen, Germany). X-ray diffraction (XRD) patterns were recorded on a Bruker D8 ADVANCE diffractometer by employing CuKa radiation. SERS signal was measured on Renishaw by Via-Reflex micro-Raman spectrometer with a 532 nm laser. The sample was exposed for 10 s at 1% laser (0.5 mW) and was scanned five times. At least 36 spots on the same SERS substrate were detected, and all collected spectra were averaged for the analysis.

### 2.3. In Situ Growth of Nanoparticles

A total of 100 mL of water was heated to about 70 °C, and 1.0 g of methyl cellulose powder was added with slow stirring. A slurry gradually formed and a uniform suspension was obtained. The suspension was then cooled to obtain a clear and transparent solution. Freshly prepared 0.01 M AgNO_3_ solution was added to the methyl cellulose solution and stirred at 20 °C for 24 h.

### 2.4. Fabrication of the Thin Film SERS Substrate

The methylcellulose solution grown in situ with AgNO_3_ was poured into the mold to make a solid film. A volume of 15 mL methyl cellulose solution was poured into a petri dish with a diameter of 60 mm and dried at 60 °C for 16 h.

### 2.5. Detection of NBA Solutions

To test the performance of the film SERS substrates, NBA was selected as the Raman model probe. Briefly, NBA of different concentrations (10^−4^ M, 10^−5^ M, 10^−6^ M, 10^−7^ M, 10^−8^ M) were prepared and the film substrate was immersed in the solution for 30 min. The film was then dried at room temperature until Raman detection.

### 2.6. Detection of Pesticide Residues

The foods (tomatoes and cucumbers) were washed with ultrapure water. To simulate SERS analysis of real samples, fruits and vegetables were peeled and the peels were cut into 1 × 1 cm^2^ squares. Then, 10 μL of the as-prepared pesticide solution was spread onto each peel square and dried. The SERS film was then wiped onto each sample and pressed. The substrate was held at a certain pressure for 10 s, before being peeled off slowly and carefully. The SERS film substrate was then moved to the glass slide for further SERS analysis.

## 3. Results and Discussion

### 3.1. Fabrication and Morphological Characterization of the MC/Ag NPs SERS Film

A stable and high-performing SERS film substrate was achieved by decorating Ag NPs in an MC matrix. Figure 1 shows the schematic diagram of preparation and detection of the SERS film substrate. The flexible film substrate showed a good ability to adapt to different shapes. It can not only be directly immersed in the analysis solution, but also be wiped on the surface of real samples to detect pesticides.

The methylcellulose solution containing silver nitrate was poured into a petri dish and dried at 60 °C. After 16 h, a reddish-brown solid film was formed, which can be seen in Figure 1A. We characterized the morphology of the film by SEM, and the results are shown in Figure 1B. The fibrous structure of the cellulose and the successful growth of the Ag NPs could be clearly seen in the methylcellulose matrix. The “hot spots” of the SERS substrate were greatly enhanced, and the Raman signal was significantly improved. As shown in Figure 1C, elemental mapping was done with SEM-EDX. From the three element distribution diagrams in Figure 1C, we can also see that the Ag NPs are evenly distributed inside the film and on its surface, which provides favorable conditions for the good sensitivity and uniformity of the substrate. It is well known that methyl cellulose has a certain degree of water solubility, but this will cause inconvenience to actual detection on site. We have characterized the hydrophilic and hydrophobic properties of methyl cellulose, and the contact angle of water droplets was 116.7° as shown in Appendix A, which proves the hydrophobic properties of the surface of the MC/Ag NPs film.

The XRD patterns of methylcellulose and methylcellulose decorated with silver nanoparticles are shown in Figure 2. The two samples showed the same diffraction peaks around 2θ at 20.9°, 31.9°, and 45.5°, which was the result of the low levels of crystalline methylcellulose. In addition, from the MC/Ag NPs film, four distinct peaks were observed at 38.2, and 77.5, which are attributed to the (1 1 1), and (3 1 1) planes, of the face-centered cubic silver crystals. The presence and contents of elements in the MC/Ag NPs film in Appendix A were also shown using energy dispersion spectroscopy (EDS).

### 3.2. Performance of the MC/Ag NPs Film

Sensitivity and uniformity are the main factors affecting any SERS substrate. Therefore, to evaluate the SERS performance, the MC/Ag NPs film substrate was used to detect NBA at different concentrations (Figure 3A). The Raman band at 590 cm^−1^ (the characteristic peak of NBA) was used for the quantitative assessment of SERS sensitivity. Even when NBA concentration was reduced to 10 nM, the SERS spectral characteristic peak of 590 cm^−1^ could still be recognized. We can also see from the figure that the NBA has a wide linear detection range, about 10^−4^ to 10^−8^ M. To further evaluate the SERS performance of our MC/Ag NPs film substrate, we calculated the enhancement factor (EF) [31] as follows.
EF = (I_SERS_/N_SERS_)/(I_NR_/N_NR_) (1)

In Equation (1), I_SERS_ is the integrated intensities of SERS, and I_NR_ is the normal Raman scattering spectra of NBA. N_SERS_ is the number of molecules in the laser excitation region on the MC/Ag NPs film. N_SERS_ and N_NR_ were derived from calculations mentioned in a previous study [32]. Therefore, The MC/Ag NPs film calculated from Equation (1) was 1.7 × 10^5^.

To further study the uniformity of the MC/Ag NPs film substrate, we separately studied the uniformity on the same substrate and the uniformity between different substrates. Thirty-six detection points were randomly selected for Raman analyses, and NBA was used as the Raman marker. The Raman intensities at 590 cm^−1^ was used as the standard to draw the bar graph in Figure 3B, and the relative standard deviation (RSD) was 5.39%. In order to evaluate the substrate-to-substrate uniformity, the SERS spectra of 40 points randomly selected from eight different substrates were measured (Figure 4A). In addition, we used the Raman intensities at 590 cm^−1^ as the standard to draw the bar graph in Figure 4B, and the RSD was 7.47%. It was generally believed that the point-to-point or substrate-to-substrate RSD of substrates for quantitative analyses should be less than 20% [33,34]. Therefore, the results show that the MC/Ag NPs film exhibited acceptable reproducibility and could be used as a reliable SERS substrate for quantitative analysis.

Stability is the key to the practical application of MC/Ag NPs film substrates [28]. However, Ag NPs will gradually be oxidized at room temperature, resulting in the reproducibility of the SERS signal of the substrate gradually decreasing. Thus, to assess the stability of the substrate, the MC/Ag NPs film was stored at room temperature for two months, after which it was immersed in freshly prepared 10^−6^ M NBA (Figure 4C). The Raman intensity of NBA at 590 cm^−1^ did not decrease significantly, and the RSD was 9.95% (Figure 4D). Thus, those results indicated that the MC/Ag NPs film had good stability at room temperature.

### 3.3. Detection of Thiram on MC/Ag NPs Films

Figure 5A shows the MC/Ag NPs film as the SERS substrate and the SERS spectrum of the thiram solution with the concentration increased from 0.024 mg/L to 240 mg/L. The Raman characteristic peaks at 1146, 1380, and 1505 cm^−1^ become more intense as the thiram concentration increases. It can also be seen that when the concentration of thiram was reduced to 0.024 mg/L, the Raman band remained clearly distinguishable. However, when the concentration of thiram is lower than 0.024 mg/L, the Raman signal cannot be monitored. Therefore, we believe that the limit of detection can reach 0.024 mg/L. In this study, the methyl cellulose film decorated with Ag NPs had more “hot spots” with increasing thiram concentrations, as more Ag NPs were linked with the thiram and then more “hot spots” take effect. Therefore, we believe that the main contribution effect of the SERS substrate was its LSPR effect. Because the Raman peak at 1380 cm^−1^ was the strongest peak in the SERS spectrum, we used this band to further study the linear function of thiram concentration. The relationship between the SERS intensity at 1380 cm^−1^ and the thiram concentration is shown in Figure 5B. In the figure, each error bar corresponds to the standard deviation of the SERS intensity collected at five points for each sample. It is observed that the Raman peak intensity of each sample is similar, which indicates the reliability of the MC/Ag NPs film substrate. It can also be seen from Figure 5B that as the logarithm of the thiram concentration increased, the intensity of the Raman spectrum increased non-linearly. When the concentration of thiram was less than 24 mg/L, an excellent linear relationship occurred, and the R^2^ of the calibration was 0.9979. When the thiram concentration was higher than 24 mg/L, the intensity of the Raman spectrum increased sharply.

### 3.4. Application in Real Samples

Thiram is a fungicide used in agriculture to protect food from fungal diseases [35]. To evaluate the practical application of the MC/Ag NPs film, the MC/Ag NPs film substrate was used to detect thiram on various peels. A total of 10 μL of thiram was sprayed on 1 cm × 1 cm squares of tomato and cucumber peels, and then dried in the air. The SERS spectra of the thiram obtained from different peels are shown in Figure 6. The thiram band can be clearly observed at 1380 cm^−1^, which is the result of C−N bond stretching. Obvious characteristic bands can also be observed at 562, 929, 11,466, and 1505 cm^−1^, which are attributed to S−S bond stretching, C=S bond stretching or C−N bond stretching, C−N bond stretching or swing methyl group mode, and C−N stretching or deformation of the methyl group [28,36]. It is well known that the surface roughness and chemical composition of the peel are different, so there are some differences in the spectral signal intensity of cucumber and tomato peels. Nevertheless, we can still detect the 2.4 ng/cm^2^ thiram content on actual samples. The limit of detection was far below the allowable maximum residue limit (MRLs) of thiram in fruits and vegetables set by relevant government agencies [37], indicating that the sensitivity of the MC/Ag NPs film was sufficient to identify thiram residues on the peel. We compare the detection limits of thiram in Table 1 with some reported flexible SERS substrates. The MC/Ag NPs film developed by us achieved good results. More importantly, the preparation process of the MC/Ag NPs film was convenient and simple, and did not involve any complex operations or expensive equipment. The MC/Ag NPs film is also highly flexible and stable.

## 4. Conclusions

In this study, we demonstrated that methylcellulose decorated with Ag NPs can be used as a new type of flexible SERS substrate. The MC/Ag NPs film can not only be directly immersed in liquid for SERS detection, but can also be wiped on irregular solid surfaces for SERS detection, thus making it possible to easily sample complex surfaces. The presence of Ag NPs provides rapid and sensitive detection activity for the MC/Ag NPs film substrate. The detection of standard SERS molecules (NBA) revealed the high sensitivity, remarkable reproducibility, and excellent stability of MC/Ag NPs films, which can be produced at low cost. As a practical application, we detected thiram pesticide residues on cucumber and tomato peels. Compared to existing technologies on the market, the proposed MC/Ag NPs film can realize the rapid detection of pesticides without any other treatment. In addition, the MC/Ag NPs film was quite stable when exposed to air and is more convenient to store and transport. Therefore, the use of MC/Ag NPs films to effectively extract targets from complex surfaces supports the use of SERS technology for real-world applications.

## Data Availability

The data presented in this study are available on request from the corresponding author.

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
