# Peer review of "In-Situ Synthesis of Methyl Cellulose Film Decorated with Silver Nanoparticles as a Flexible Surface-Enhanced Raman Substrate for the Rapid Detection of Pesticide Residues in Fruits and Vegetables"

_materials, 2021, doi:10.3390/ma14195750_

Round 1
Reviewer 1 Report
A new flexible SERS substrate have been described and used for detection of a pesticide on tomato and cucumber peels. The substrate is noteworthy for practical application because it shows good properties such as uniformity, sensitivity and stability. Moreover, it is easy to prepare and costs of its preparation are relatively low. The new substrate has been thoroughly investigated. Its good detection performance for a thiram pesticide is compared with other flexible substrates described in the literature. However, it is not clear how a simple “stick and peel” procedure can influence on the amount of pesticide transferred from the peel. It seems to me this amount (and thus sensitivity) should be dependent on experimentalist and especially his will to show that he has prepared a remarkable substrate.
In my opinion the paper can be accepted for publication in Materials, nevertheless some additional comments and corrections are necessary.
- Page 3, point 2.1. We read that “Fruits and vegetables such as plums, tomatoes, and cucumbers were bought ….” However, only investigations of tomatoes and cucumbers are described in the manuscript.
- Page 5, Figure 3C . It is not clear what show four SEM-EDX images. An additional description is necessary.
- Page 6, point 3.2. I wonder why information about the enhancement factor is transferred to Supporting Information. This is so important factor that it should be described in the main text.
- Page 6, point 3.3. The first section we read about the data which are already described in the previous point 3.2. Such repetition should be avoided, therefore, it seems to me that the points 3.2 and 3.3 should be merged into one point.
- Page 7, Figures 4 and 5. In the panels signed as A we see a collection of SERS spectra recorded at 36 random sites of substrates. In my opinion these figures and unreadable and show nothing important, therefore that they should be removed. Presentation of panels B together with a reference to SERS spectra in Figure 3 should be enough.
- Page 8, the first sentence. We see Figure 5A but it should be Figure 6A.
- Page 8, line 10 from above. What does mean “LSPR effect”?
- Page 8, Figure 6. I think that the inset in the panel B should be removed. Because of small letters and numbers it is difficult to read. Moreover, this is an unnecessary repetition of the plot distinguished by a red frame.
- Page 9. I wonder whether Table 1 is necessary. It shows the assignments of Raman bands of thiram pesticide on basis of Refs [41, 42] but the same description we can find in the text. I suggest to remove Table 1 and eventually extend the description in the text.
- Page 10, Figure 7. In the figure caption we read that the panels A and B show tomato and cucumber peels, respectively, but it is vice versa. Moreover, in the inset of panel A we see a combination of one cucumber and four small tomatoes.
- I have found quite many linguistic mistakes. It is necessary to correct the text !
Reviewer 2 Report
The present paper is very nice, original and well presented. I have only some remarks and therefore I suggest to publish it after minor revisions.
1) English style should be carefully revised.
2) The auhtors should add some data about the thickness of the MC foils prepared and discuss the thickness influence onto silver nanoparticles growth and detection effeciency of pesticides.
3) MC foils: the authors should explain if the foils prepared can be re-used in order to understand the sustainability of the process here proposed.
Reviewer 3 Report
Review of the article “In-situ synthesis of methyl cellulose film decorated with silver nanoparticles as a flexible surface-enhanced Raman substrate for rapid detection of pesticide residues in fruits and vegetables” by Zhang et al.
The authors describe the development, testing and application of a flexible substrate (methyl-cellulose film doped with silver nano-particles) as a sampling medium to be used in the spectroscopic (SERS) detection of pesticide residue in fluits and vegetables. The produced film is characterized using SEM (surface morphology), EDS and XRD. Then, the film is calibrated in the detection of Nile Blue A and its sensitivity as well as the spatial uniformity of the signal are determined. Finally, the films are used to sample surfaces of fruits/vegetables and determine the presence of a common pesticide (Thiram). The sensitivity of the film in the context of SERS is found to be lower than the safely limits required by regulating agencies. It is also better than several previously reported studies.
This is a well-conducted study with a clear positive result. The paper is well-written and the key points satisfactorily addressed. It is suitable for publication in “Materials”. If possible, I would recommend the authors address the issue “why their film is superior to earlier results”. Also a few minor editorial points:
Second page, start of last paragraph, remove “2” from “and2”
Top of page 8, I think it should be “Figure 6” instead of “Figure 5”
Page 9, rephrase “limit f detection limit” to something that reads better
